# Post kala azar dermal leishmaniasis and leprosy prevalence and distribution in the Muzaffarpur health and demographic surveillance site

Epco Hasker[1☯]*, Paritosh Malaviya[2☯], Vivek Kumar Scholar[2,3], Pieter de Koning[1], Om Prakash Singh[2], Sangeeta Kansal[2], Kristien Cloots[1], Marleen Boelaert[1], Shyam Sundar[2]

1 Institute of Tropical Medicine, Antwerp, Belgium, 2 Department of Medicine, Institute of Medical Sciences, Banaras Hindu University, Varanasi, India, 3 Kala-azar Medical Research Center, Muzaffarpur, India

☯ These authors contributed equally to this work.
* ehasker@itg.be

**Data Availability Statement:** The data supporting the findings of this study/publication are retained at the Institute of Tropical Medicine, Antwerp and will

## Abstract

### Introduction

Post-kala-azar dermal leishmaniasis (PKDL) is a skin manifestation that is a late clinical outcome of visceral leishmaniasis (VL). Its presentation is similar to leprosy, and the differential diagnosis is not always easy. In VL endemic rural areas of Bihar, India, both infectious diseases co-exist. This observational study aimed to determine the prevalence and distribution of both conditions in an area that had until recently been highly endemic for VL.

### Methods

We conducted a door-to-door survey in an area that belongs to the Health and Demographic Surveillance Site (HDSS) of Muzaffarpur, Bihar, India. Within the HDSS we selected the villages that had reported the highest numbers of VL cases in preceding years. All consenting household members were screened for skin conditions, and minor conditions were treated on the spot. Upon completion of screening activities at the level of a few villages, a dermatology clinic ("skin camp") was conducted to which suspect leprosy and PKDL patients and other patients with skin conditions requiring expert advice were referred. We studied the association between distance from an index case of leprosy and the probability of disease in the neighborhood by fitting a Poisson model.

### Results

We recorded a population of 33,319, out of which 25,686 (77.1%) were clinically screened. Participation in skin camps was excellent. Most common conditions were fungal infections, eczema, and scabies. There were three PKDL patients and 44 active leprosy patients, equivalent to a prevalence rate of leprosy of 17.1 per 10,000. Two out of three PKDL patients had a history of VL. Leprosy patients were widely spread across villages, but within villages, we found strong spatial clustering, with incidence rate ratios of 6.3 (95% C.I. 1.9–21.0) for household members and 3.6 (95% C.I. 1.3–10.2) for neighbors within 25 meters, with those living at more than 100 meters as the reference category.

not be made openly accessible due to ethical and privacy concerns. Data can however me made available after approval of a motivated and written request to the Institute of Tropical Medicine at ITMresearchdataaccess@itg.be.

**Funding:** This work was supported by the Extramural Program of the National Institute of Allergy and Infectious Disease (NIAID), National Institute of Health (TMRC Grant No. U19AI074321). The funders had no role in study design, data collection and analysis, decision to publish, or preparation of the manuscript.

**Competing interests:** The authors have declared that no competing interests exist.

## Discussion

Even in this previously highly VL endemic area, PKDL is a rare condition. Nevertheless, even a single case can trigger a new VL outbreak. Leprosy is also a rare disease, but current prevalence is over 17 times the elimination threshold proclaimed by WHO. Both diseases require continued surveillance. Active case finding for leprosy can be recommended among household members and close neighbors of leprosy patients but would not be feasible for entire populations. Periodic skin camps may be a feasible and affordable alternative.

## Author summary

We describe a survey for post kala azar dermal leishmaniasis (PKDL) and leprosy carried out in a surveillance site in Bihar, India, that has until recently been highly endemic for visceral leishmaniasis (VL). Both leprosy and VL are subject to elimination initiatives, for both diseases the target prevalence is set at less than 1 case per 10,000 population. PKDL, a dermatological sequel of VL, is important because it can act as a reservoir for VL in inter-epidemic periods. So far very little is known about prevalence of PKDL in India and the frequency of PKDL among former VL patients. Our survey was population based, allowing us to assess PKDL prevalence not only among former VL patients but also among those not known to have suffered from VL in the past. We conducted door-to-door screening and suspect cases of leprosy and PKDL were referred to a 'skin camp', an outreach consultation for skin conditions at village level. We recorded geographic coordinates of each household screened, thus we were able to explore spatial associations. Participation in the survey and in the skin camps was very good. On a population of over 25,000 screened we found only three PKDL cases but 44 active leprosy patients. For leprosy this equates to a prevalence of 17.1 per 10,000, way above the elimination threshold. Of our three PKDL cases, two had occurred in known former VL cases, one was in a person who reported never to have suffered from VL. For leprosy we found strong spatial associations with a six-fold increase in risk for household members but also a three to four-fold increase for neighbors at less than 25 meters. The three PKDL cases were from three different villages, their numbers were too low for any meaningful spatial analysis. We conclude that leprosy is still a major problem in the area that requires more efforts. Active screening of household contacts is rational but should be extended to nearby neighbors. Though relatively efficient, this approach would miss out on the bulk of leprosy patients. PKDL is a rare condition but needs to be kept under surveillance. Skin camps can provide a feasible alternative to door-to-door screening, allowing to target both conditions simultaneously.

## Introduction

Post-kala-azar dermal leishmaniasis (PKDL) is a late clinical outcome of visceral leishmaniasis (VL). In East Africa it is reported to occur among up to 50% of all VL patients, on the Indian Subcontinent (ISC) it is less common.[1,2] PKDL may present as a macular, a maculopapular, or a nodular rash. It is usually self-limiting and causes only minor inconvenience to the patient. The importance of PKDL lies in the fact that PKDL lesions are infectious to sand flies and may thus sustain transmission of *L.donovani*, acting as inter-epidemic reservoirs.[3] This

becomes increasingly important in the context of the VL elimination initiative on the ISC which aims to bring down the annual incidence of VL at sub-district level (a population of approximately 100,000) to below 1 per 10,000 population by the end of 2015.[4] This target date has been extended to 2017 and more recently to 2020.

Based on the clustering of cases in space and time, VL transmission in the ISC is assumed to be driven by the presence of clinical VL cases in the community. The re-emergence of VL in India after the malaria eradication campaigns of the nineteen fifties and sixties was attributed to PKDL cases.[5] In such a low prevalence context, obviously the smaller the number of clinical VL patients, the higher the potential share of PKDL in transmission. Although as was stated earlier, PKDL is assumed to be less common on the Indian sub-continent than in East Africa, there are regional differences and in India in particular very few studies have been done.[6,7] In the Muzaffarpur Health and Demographic Surveillance System (HDSS), situated in a VL endemic area in Bihar, India, comprising of 50 contiguous villages and with a population of approximately 90,000, 339 new VL cases were reported from 2007 to 2018, but only 5 cases of PKDL. Considerable underreporting of PKDL is suspected.

An important condition in the differential diagnosis of PKDL is leprosy.[1] Just like VL, leprosy is among the neglected tropical diseases targeted for elimination by the World Health organization (WHO).[8] The global target of a prevalence of less than 1 case per 10,000 population was achieved by 2000. By 2005 the target was also reached at country level, albeit with a few exceptions. India was among the countries that by 2005 had achieved an overall leprosy prevalence of less than 1 per 10,000, an enormous reduction from the 57.8 per 10,000 observed in 1983.[9] However, much of this reduction in prevalence was due to changes in duration of treatment and case definitions. When multi-drug therapy (MDT) was introduced in 1982, treatment duration–and hence the period during which a patient is considered a case—was reduced from lifelong to two years at most. This led to a spectacular decrease in prevalence rates of leprosy worldwide. Towards the end of the 1990s the duration of leprosy treatment was again reduced to a maximum of one year, once again significantly reducing prevalence. However, the reported annual number of new cases of leprosy in India (the incidence rate) has remained fairly constant since 2006, fluctuating between 139,252 and 126,800 without much of a noticeable trend.[10] The states of Bihar, Maharashtra and West Bengal, which had earlier achieved the elimination threshold, were again above the threshold at the end of 2013.[11]

Based on mathematical models, Blok *et al*. predict that leprosy is likely to remain a problem in highly endemic regions.[12] In 2010, 2,177 leprosy cases were detected among 15 million screened in a national sample survey in India, equivalent to a weighted prevalence of 2.5 per 10,000.[13] Among those, 11.2% had visible deformities (WHO grade 2 disability), indicative of longstanding disease. The fact that the leprosy is now considered eliminated from India as the target has been met, in all probability means that less resources are made available, less case finding activities are conducted, and therefore the incidence does not change. To examine this hypothesis, more data on leprosy at population level are needed.

Our study aimed to assess the prevalence of PKDL and leprosy in villages of Bihar State that have been endemic for VL until very recently. We also studied the spatial distribution of cases of either disease, to further our understanding of the epidemiology of both infectious diseases but also to examine potential synergies in the response measures taken to control them.

## Methods

### Study population

As stated earlier, the Muzaffarpur-TMRC Health and Demographic Surveillance System (HDSS) is a demographic surveillance system covering a population of over 90,000 in 50

villages of a VL endemic district in Bihar, India.[14] In this area, exhaustive household surveys are organized on an annual basis to record demographic data. Heads of all households are interviewed about births, deaths, migrations in, migrations out, and cases of VL and PKDL in the past 12 months. Each household has been georeferenced. From this area, we selected the 27 villages that had reported the highest numbers of VL cases (276 out of 339) in the 10 years preceding the survey. The five PKDL cases mentioned earlier had all occurred in these villages.

## Study procedures

Starting early 2017 and until mid-2018, we conducted a door-to-door survey for skin conditions. Trained staff examined each household member for the presence or absence of skin lesions and recorded the location of lesions on a pre-printed form featuring a body map. Those with skin lesions were referred to a dermatology clinic ("skin camp") which was conducted each time screening activities were completed in up to three adjacent villages. We paid special attention to suspect cases of either PKDL or leprosy. Such patients were re-visited in their homes on the day before the skin camp to encourage their attending to the clinic. In case they did not attend they were re-visited in their homes for diagnostic confirmation.

The skin clinics were conducted by two physicians, one from a leprosy hospital and one from the KAMRC treatment center for VL and PKDL. They examined all patients and offered free treatment and care for PKDL and leprosy. PKDL was confirmed by demonstration of parasites by microscopy in a skin smear or biopsy or by qPCR. The diagnosis of leprosy was clinical. Free treatment was also provided for minor treatable skin conditions such as bacterial infections, fungal or parasitic infections and eczema.

## Data analysis

Prevalence rates of PKDL and leprosy were calculated with current cases of each disease as numerator and total population screened as denominator. We then determined for each individual, irrespective of whether or not the person was himself a case of PKDL or leprosy, the distance to the nearest (other) PKDL or leprosy case. This was done using the distance matrix module in Quantum GIS version 3.4. Distances to the nearest case were subdivided into 6 classes: household contacts and five categories of 'neighbors': at less than 25 meters, 25–50 meters, 50–75 meters, 75–100 meters, and more than 100 meters. We then fitted a Poisson model with village as random effect in Stata 14.1. We calculated prevalence rate ratios with those living at more than 100 meters from the nearest case as reference category. As offset we considered the entire (log) population, irrespective of whether or not they had participated in the screening.

## Ethical considerations

Ethical clearance for the study protocol was obtained both in Belgium (from the institutional review board of the Institute of Tropical Medicine and the Ethics Committee of the Antwerp University hospital) and in India (from the Ethics Committee of Banaras Hindu University). All subjects screened provided written informed consent; in case of illiterate subjects, a thumb print plus a signature of an independent witness were used. For minors under the age of 18, written informed consent was obtained from a parent or guardian. Informed consent procedures were approved by the respective review boards.

## Results

We recorded a total population of 33,319, of which 54.5% were female. The median age was 18 years, interquartile range from 7 to 38. The teams were able to screen 25,686 individuals, i.e.,

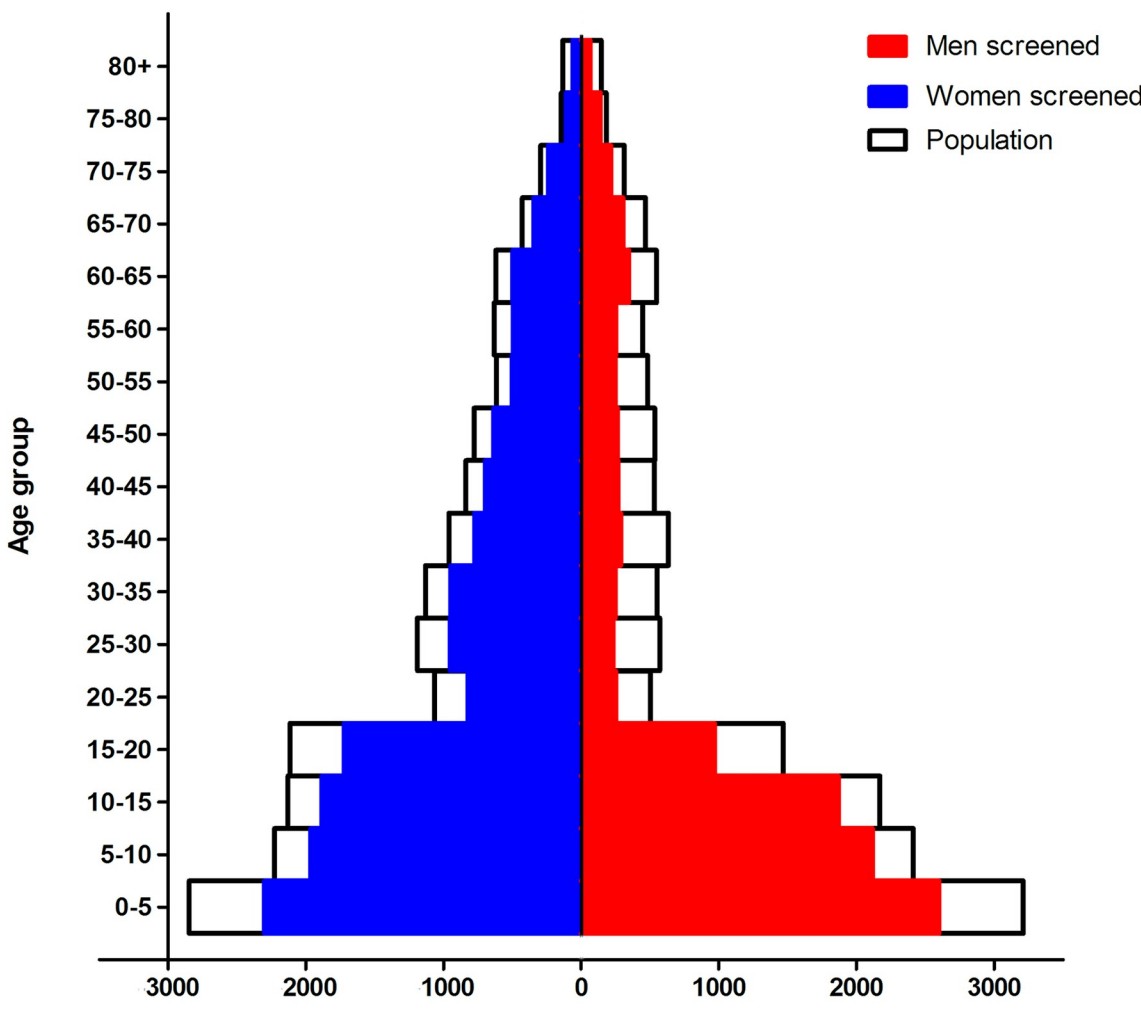

**Fig 1. Population pyramid.** Population recorded (n = 33,319) and population screened (n = 25,686).

77.1% of the population. With a few exceptions, those not screened were not present at the time of the visit. Men were underrepresented, making up only 41.7% of the population screened; in the age group above 20 years, only 30.7% were men. Part of the underrepresentation of adult men in our study population is probably due to out-migration from the region for labor reasons.[14] As can be seen from Fig 1 below, there is a major gap in the population pyramid of the Muzaffarpur HDSS among young adult men. For women, the gap is also there but less prominent.

Among those screened, 2,466 (9.6%) had skin lesions. Most common causes were fungal infections (1,075), eczema (133) and scabies (65). There were four PKDL suspects and 116 leprosy suspects. The skin camps were well attended, altogether there were 2,505 consultations in 15 camps (average of 167). All PKDL suspects and 103 leprosy suspects presented for confirmation at the skin camps. Out of the four PKDL suspects, three were confirmed, equivalent to a prevalence rate of 1.2 per 10,000. One had been diagnosed as a VL case in 2008, another one had been diagnosed with VL before the start of the HDSS (prior to 2007), the third one did not have a history of VL. Out of 103 leprosy suspects examined in the skin camps 71 were

confirmed as leprosy cases, but only 42 still had active disease. Two more were identified during the camp, bringing the total number of prevalent active leprosy cases to 44 (22 multi bacillary, 22 pauci bacillary) on a population of 25,686 screened, i.e. a prevalence rate of 17.1 per 10,000.

Fig 2 below shows the area surveyed and the locations of households in which PKDL and (active) leprosy cases were found.

The three PKDL cases were all from different villages. Given their small numbers, a further spatial analysis was not considered useful. Among leprosy patients, strong spatial clustering was apparent at neighborhood level. Four out of 44 cases (9.1%) were found among 402 persons living in the same household as a leprosy patient, who make up 1.2% of the population. Among the neighbors at less than 25 meters of an active leprosy case, who make up 3.5% of the total population, 15.9% of the leprosy patients were found. At a distance of 50 meters or more from a leprosy case, the clear excess of leprosy patients no longer seemed to be present. Details are presented in Table 1.

The apparent gradient was confirmed in the Poisson model with village as random effect. With those living at more than 100 meters as reference category we found rate ratios of 6.3 (95% CI 1.9–21.0) and 3.6 (95% CI 1.3–10.2) respectively for those living in the same household and neighbors at less than 25 meters of a leprosy patient. Beyond 25 meters differences observed were no longer statistically significant (Table 2)

## Discussion

In this poor rural area in Bihar, India, which had until recently been highly endemic for VL, we found a very low prevalence of PKDL (1.1 per 10,000). An earlier study by Singh *et al*. in 2010, targeting different villages in the same district, had identified five confirmed PKDL cases on a population of 11,466, equivalent to a prevalence of 4.4 per 10,000.[7] In addition there were four probable PKDL cases which–if included- increased the prevalence to 7.8 per 10,000. The PKDL prevalence in our current study was lower but this might be explained by the fact that more time had elapsed since the peak of the VL epidemic in the area, which was in 2007– 2008.[15] Another factor might be that the more intense surveillance within the HDSS led to earlier detection and more effective treatment of VL, with a beneficial impact on the number of PKDL cases. The area might therefore not be entirely representative in this respect for the greater Bihar. Indeed, despite continued surveillance only five new PKDL cases had been reported from the HDSS area since 2007. However, such reporting was just based on interviewing heads of households, without actual visual inspection of the skin. It is likely that some PKDL cases might have been missed during this period. A limitation of our current survey is the fact that young adult men are under represented.

Leprosy was much more prevalent, with a prevalence rate of 17.1 per 10,000, i.e. 17 times the elimination target. The observed prevalence was much higher still than the 1.5/10,000 (2,161 cases on a population screened of 14,725,525) observed by Katoch *et al*. who studied leprosy prevalence in India at national level in 2011.[16] In all likelihood this is explained by the fact that the HDSS site is an impoverished rural area in one of the poorer states of India. As had already been shown by Boelaert *et al*. the VL endemic areas of Muzaffarpur district belong to the poorest of the poor.[17] In such areas, leprosy appears far from eliminated and continued efforts are required.

Even though both incidence and prevalence of PKDL appear to be low, as has been demonstrated in West Bengal, in a non-immune population there is always a risk for VL to flare up again.[5] A few untreated PKDL patients in a non-immune population can be sufficient to trigger a new VL outbreak. Active case finding through door-to-door screening is probably the

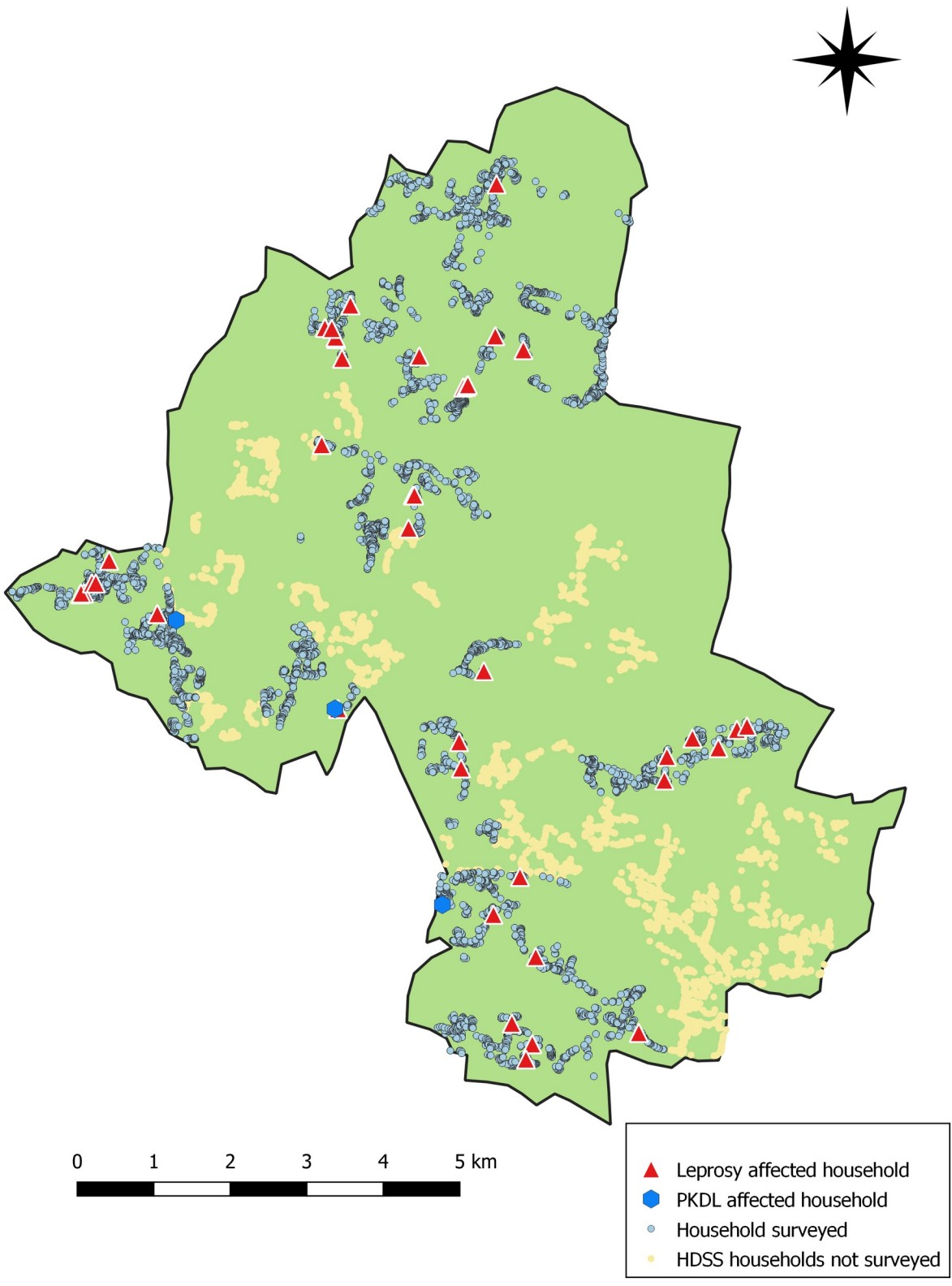

**Fig 2. Muzaffarpur HDSS.** Distribution of PKDL and leprosy in the area surveyed (map created in QGIS version 3.4.6-Madeira).

**Table 1. Distribution of leprosy patients as a function of distance to nearest (other) leprosy patient.**

| Distance to nearest (other) leprosy case | Number of leprosy patients (%) | Total Population (%) |
|---|---|---|
| Same household | 4 (9.1) | 402 (1.2) |
| Neighbor at <25 meter | 7 (15.9) | 1,167 (3.5) |
| Living at 25–50 meter | 4 (9.1) | 1,374 (4.1) |
| Living at 50–75 meter | 2 (4.6) | 1,642 (4.9) |
| Living at 75–100 meter | 2 (4.6) | 1,175 (3.5) |
| Living at >100 meter | 25 (56.8) | 27,546 (82.7) |
| Total | 44 | 33,306 |

most effective way to find the few hidden PKDL cases but unfortunately will never be sustainable. Our teams spent more than one year, with interruptions, to screen a population of approximately 33,000 and were able to examine only three-quarters of those. For leprosy, door-to-door screening could be considered when focusing on the immediate neighborhood of households in which cases have been identified. The increased risk for those living within 25 meters of a leprosy affected household found in our study confirms the observations of Moura *et al.* from Brazil who found equally high prevalence rates in households of neighbors of leprosy patients as in the patient households.[18] As Moura *et al.* also recommend it would be meaningful to extend active case finding to neighborhood contacts. Though fairly cost effective, such an approach would have missed 33 out of 44 (75%) leprosy patients identified in our survey if one were to adopt a maximum perimeter of 25 meter.

To address both leprosy and PKDL, skin camps may be a reasonable compromise between door-to-door screening and passive case finding. The skin camps in our study were very well attended. Most attendees, however, do not attend with complaints caused by PKDL or leprosy but because of other much more prevalent skin conditions. Many of those can be treated with relatively inexpensive means. In an earlier study in this district, we observed that people incur substantial costs while searching health care for VL.[19] For PKDL Garapati *et al.* also documented very long delays.[20] Responding to the basic health care needs of the population and seizing the opportunity to also search for PKDL and leprosy at the same time could be the most efficient way forward. Another important element would be to make good use of the networks of Accredited Social and Health Activists (ASHAs) as the link between the community and the health care system.[21]

## Conclusion

PKDL is a rare condition in India requiring more attention because of its potential to trigger new VL epidemics. Leprosy is far from eliminated and requires intensified case finding approaches. Targeting active case finding for leprosy to household members of cases and

**Table 2. Probability of being diagnosed with leprosy as a function of distance to the nearest index case (controlled for village of residence).**

| Distance to nearest (other) leprosy case | Rate ratio (95% confidence interval) |
|---|---|
| Same household | 6.3 (1.9–21.0) |
| Neighbor at <25 meter | 3.6 (1.3–10.2) |
| Living at 25–50 meter | 1.9 (0.6–6.2) |
| Living at 50–75 meter | 0.9 (0.2–4.0) |
| Living at 75–100 meter | 1.2 (0.2–5.4) |
| Living at >100 meter | Ref. |

neighbors seems rational, but would miss out on a large number of cases. A combination of targeted active case finding for leprosy, with screening for PKDL in hotspots of previous transmission, and regular skin camps seems the most realistic approach to tackle both diseases. For PKDL this should be combined with active follow-up of previously treated VL cases. A more in-depth cost-effectiveness study from the health system perspective would be welcome.

## Supporting information

**S1 Checklist. STROBE checklist.**
(DOCX)

## Acknowledgments

We wish to acknowledge Dr. Subhash Suman from the TLM Muzaffarpur leprosy hospital and Dr. Deepak Kumar from the Kala azar medical research center (KAMRC) in Muzaffarpur, in particular for their contributions to the success of the skin camps.

## Author Contributions

**Conceptualization:** Epco Hasker, Paritosh Malaviya, Pieter de Koning, Om Prakash Singh, Sangeeta Kansal, Kristien Cloots, Marleen Boelaert, Shyam Sundar.

**Data curation:** Epco Hasker, Vivek Kumar Scholar.

**Formal analysis:** Epco Hasker, Paritosh Malaviya, Kristien Cloots, Marleen Boelaert.

**Investigation:** Vivek Kumar Scholar, Pieter de Koning, Om Prakash Singh.

**Methodology:** Epco Hasker, Pieter de Koning, Om Prakash Singh, Shyam Sundar.

**Project administration:** Om Prakash Singh.

**Software:** Paritosh Malaviya.

**Supervision:** Paritosh Malaviya, Vivek Kumar Scholar, Pieter de Koning, Sangeeta Kansal, Marleen Boelaert.

**Validation:** Kristien Cloots.

**Writing – original draft:** Epco Hasker.

**Writing – review & editing:** Paritosh Malaviya, Vivek Kumar Scholar, Pieter de Koning, Om Prakash Singh, Sangeeta Kansal, Kristien Cloots, Marleen Boelaert, Shyam Sundar.

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
