## [Decision Letter · Decision Letter 0]

20 Aug 2019

Dear Dr. Hasker:

Thank you very much for submitting your manuscript "Post kala azar dermal leishmaniasis and leprosy prevalence and distribution in the Muzaffarpur health and demographic surveillance site" (PNTD-D-19-01032) for review by PLOS Neglected Tropical Diseases. Your manuscript was fully evaluated at the editorial level and by independent peer reviewers. The reviewers appreciated the attention to an important topic but identified some aspects of the manuscript that should be improved.

We therefore ask you to modify the manuscript according to the review recommendations before we can consider your manuscript for acceptance. Your revisions should address the specific points made by each reviewer.

(1) A letter containing a detailed list of your responses to the review comments and a description of the changes you have made in the manuscript.

(2) Two versions of the manuscript: one with either highlights or tracked changes denoting where the text has been changed (uploaded as a "Revised Article with Changes Highlighted" file ); the other a clean version (uploaded as the article file).

(3) If available, a striking still image (a new image if one is available or an existing one from within your manuscript). If your manuscript is accepted for publication, this image may be featured on our website. Images should ideally be high resolution, eye-catching, single panel images; where one is available, please use 'add file' at the time of resubmission and select 'striking image' as the file type. 

Please provide a short caption, including credits, uploaded as a separate "Other" file. If your image is from someone other than yourself, please ensure that the artist has read and agreed to the terms and conditions of the Creative Commons Attribution License at http://journals.plos.org/plosntds/s/content-license (NOTE: we cannot publish copyrighted images). 

(4) Appropriate Figure Files 

Please remove all name and figure # text from your figure files upon submitting your revision. Please also take this time to check that your figures are of high resolution, which will improve both the editorial review process and help expedite your manuscript's publication should it be accepted. Please note that figures must have been originally created at 300dpi or higher. Do not manually increase the resolution of your files. For instructions on how to properly obtain high quality images, please review our Figure Guidelines, with examples at: http://journals.plos.org/plosntds/s/figures

While revising your submission, please upload your figure files to the Preflight Analysis and Conversion Engine (PACE) digital diagnostic tool, https://pacev2.apexcovantage.com/ PACE helps ensure that figures meet PLOS requirements. To use PACE, you must first register as a user. Then, login and navigate to the UPLOAD tab, where you will find detailed instructions on how to use the tool. If you encounter any issues or have any questions when using PACE, please email us at figures@plos.org.

We hope to receive your revised manuscript by Oct 19 2019 11:59PM. If you anticipate any delay in its return, we ask that you let us know the expected resubmission date by replying to this email.

To submit your revised files, please log in to https://www.editorialmanager.com/pntd/

Sincerely,

Fabiano Oliveira

Guest Editor

Jesus Valenzuela

Deputy Editor

Reviewer's Responses to Questions

**Key Review Criteria Required for Acceptance?**

**Methods**

-Are the objectives of the study clearly articulated with a clear testable hypothesis stated?

-Is the study design appropriate to address the stated objectives?

-Is the population clearly described and appropriate for the hypothesis being tested?

-Is the sample size sufficient to ensure adequate power to address the hypothesis being tested?

-Were correct statistical analysis used to support conclusions?

-Are there concerns about ethical or regulatory requirements being met?

Reviewer #1: Study objectives, methods, and ethics are clearly described. No comments.

Reviewer #2: Line 78 -79 is repeated in line 93-94

**Results**

-Does the analysis presented match the analysis plan?

-Are the results clearly and completely presented?

-Are the figures (Tables, Images) of sufficient quality for clarity?

Reviewer #1: Study results are clearly presented, and matching the analysis plan. No comments.

Reviewer #2: Line 201: Only 4 PKDL suspects – this is quite a low number traditionally from ACD programmes that target PKDL, but may reflect the expertise of the physician and/or the lack of misdiagnosis with leprosy.

**Conclusions**

-Are the conclusions supported by the data presented?

-Are the limitations of analysis clearly described?

-Do the authors discuss how these data can be helpful to advance our understanding of the topic under study?

-Is public health relevance addressed?

Reviewer #1: Authors fail to discuss the possible biases in the prevalence estimates, due to sub-optimal performance of the diagnostic confirmation tests used, or to the significant underreporting of adult males in the survey. This may have resulted in an underestimation of the prevalence estimates.

Reviewer #2: (No Response)

**Editorial and Data Presentation Modifications?**

Reviewer #1: (No Response)

Reviewer #2: Out of interest were the 276 cases of previous VL found during the survey?

**Summary and General Comments**

Reviewer #1: I would like to congratulate the authors with this excellent study. This community survey elucidates the current prevalence of two important skin diseases, which are subject to elimination, and based on a spatial analysis suggests options for case finding strategies to support elimination.

Although I fully agree with the conclusion that a more in-depth cost-effectiveness study from the health system perspective is needed, the discussion on possible case finding strategies remain a bit thin. 

Authors suggest skin camps as an efficient and effective case finding strategy, however, only less than 2% of the people attending the skin camps were diagnosed with either leprosy or PKDL. Skin camps are a very resource intensive investment, which would have to be organised with a regular frequency in order to pick up the relatively low number of PKDL and leprosy cases. Should this be a priority in resource allocation for public health in a state, which is suffering from dire poverty and more important public health problems. Moreover, individuals with mild (macular) PKDL lesions may chose not to attend the skin camps, and may decline the treatment, which is long and with significant side effects. 

Authors suggest using the ASHAs network in case finding, but do not discuss what role these community health workers could play in active case finding strategies and to what extent they can make clinical (suspect) diagnoses, or how an additional task for skin diseases would compete with other priority tasks in the workload of the ASHAs.

In view of the clear case clustering for leprosy, a strategy of door-to-door screening in the immediate neighbourhood of identified cases could be cost-effective, considering the relatively high prevalence of the disease. However, for PKDL I would like to challenge authors to conclude that there may not be a cost-effective and sustainable case finding strategy, and that this has implications for sustainable VL elimination.

Reviewer #2: Very clear and concisely written paper. Its true that the main limitation of this is validity in other settings that are not so well controlled, but at the same time the incidence and pathophysiological evolution of PKDL should not be much different to elsewhere.

Well done!

PLOS authors have the option to publish the peer review history of their article (what does this mean?). If published, this will include your full peer review and any attached files.

Reviewer #1: No

Reviewer #2: No

---

## [Editor Report · Decision Letter 1]

18 Sep 2019

Dear Dr. Hasker,

We are pleased to inform you that your manuscript, "Post kala azar dermal leishmaniasis and leprosy prevalence and distribution in the Muzaffarpur health and demographic surveillance site", has been editorially accepted for publication at PLOS Neglected Tropical Diseases.

Before your manuscript can be formally accepted and sent to production you will need to complete our formatting changes, which you will receive in a follow up email. Please note: your manuscript will not be scheduled for publication until you have made the required changes.

IMPORTANT NOTES

* Copyediting and Author Proofs: To ensure prompt publication, your manuscript will NOT be subject to detailed copyediting and you will NOT receive a typeset proof for review. The corresponding author will have one final opportunity to correct any errors when sent the requests mentioned above. Please review this version of your manuscript for any errors.

* If you or your institution will be preparing press materials for this manuscript, please inform our press team in advance at plosntds@plos.org. If you need to know your paper's publication date for media purposes, you must coordinate with our press team, and your manuscript will remain under a strict press embargo until the publication date and time. PLOS NTDs may choose to issue a press release for your article. If there is anything that the journal should know, please get in touch.

*Now that your manuscript has been provisionally accepted, please log into EM and update your profile. Go to http://www.editorialmanager.com/pntd, log in, and click on the "Update My Information" link at the top of the page. Please update your user information to ensure an efficient production and billing process.

*Note to LaTeX users only - Our staff will ask you to upload a TEX file in addition to the PDF before the paper can be sent to typesetting, so please carefully review our Latex Guidelines [http://www.plosntds.org/static/latexGuidelines.action] in the meantime.

Best regards,

Fabiano Oliveira

Guest Editor

Jesus Valenzuela

Deputy Editor

---

## [Editor Report · Acceptance letter]

21 Oct 2019

Dear Dr. Hasker,

We are delighted to inform you that your manuscript, "Post kala azar dermal leishmaniasis and leprosy prevalence and distribution in the Muzaffarpur health and demographic surveillance site," has been formally accepted for publication in PLOS Neglected Tropical Diseases.

Best regards,

Serap Aksoy

Editor-in-Chief

Shaden Kamhawi

Editor-in-Chief
